# Molecular transport through large-diameter DNA nanopores

Swati Krishnan[1,2,*], Daniela Ziegler[1,2,*], Vera Arnaut[1,2,*], Thomas G. Martin[2,3,†], Korbinian Kapsner[1,2], Katharina Henneberg[4], Andreas R. Bausch[4], Hendrik Dietz[2,3] & Friedrich C. Simmel[1,2,*]

DNA-based nanopores are synthetic biomolecular membrane pores, whose geometry and chemical functionality can be tuned using the tools of DNA nanotechnology, making them promising molecular devices for applications in single-molecule biosensing and synthetic biology. Here we introduce a large DNA membrane channel with an $\approx 4\,nm$ diameter pore, which has stable electrical properties and spontaneously inserts into flat lipid bilayer membranes. Membrane incorporation is facilitated by a large number of hydrophobic functionalizations or, alternatively, streptavidin linkages between biotinylated channels and lipids. The channel displays an Ohmic conductance of $\approx 3\,nS$, consistent with its size, and allows electrically driven translocation of single-stranded and double-stranded DNA analytes. Using confocal microscopy and a dye influx assay, we demonstrate the spontaneous formation of membrane pores in giant unilamellar vesicles. Pores can be created both in an outside-in and an inside-out configuration.

[1] Physik-Department E14, Technische Universität München, Am Coulombwall 4a, 85748 Garching, Germany. [2] Zentrum für Nanotechnologie und Nanomaterialien/WSI, Technische Universität München, Am Coulombwall 4a, 85748 Garching, Germany. [3] Physik-Department E69, Technische Universität München, Am Coulombwall 4a, 85748 Garching, Germany. [4] Physik-Department E27, Technische Universität München, James-Franck-Straße 1, 85748 Garching, Germany. † Present address: Medical Research Council - Laboratory of Molecular Biology, Francis Crick Avenue, Cambridge Biomedical Campus, Cambridge CB2 0QH, U.K. * These authors contributed equally to this work. Correspondence and requests for materials should be addressed to F.C.S. (email: simmel@tum.de).

n the past few years, DNA-based nanostructures[1–3] have been developed that mimic the shape and function of naturally occurring membrane channels[4–6]. They have been shown to incorporate into lipid bilayer membranes and create pores with an electrical conductance in the nanosiemens range. In addition, DNA origami structures have been used as biomolecular adaptors for solid-state nanopores[5,7,8]. The geometry and chemical functionality of this novel class of artificial nanopores can be rationally designed using computer-aided molecular engineering tools that are available for DNA nanotechnology[9]. In contrast to solid-state nanopores[10], DNA-based pores can be produced with molecular precision and chemically functionalized via incorporation of appropriate DNA bioconjugates. Protein or peptide membrane pores also offer molecularly defined dimensions; however, their geometry cannot be modified as easily as for DNA nanostructures, and their chemical functionalization typically is more cumbersome. These features make DNA-based membrane channels highly promising biomolecular devices for applications in single-molecule biosensing[10–18] or drug delivery, or as components for artificial cells[19,20]. First-generation DNA membrane pores had a low tendency to incorporate into lipid bilayers, and often displayed unstable conductance values and undesired channel gating[4].

In the following, we address these issues by developing a series of nanopore structures, for which we vary scaffold routing and shape as well as the number of hydrophobic modifications. These studies suggest that a larger membrane contact area and a correspondingly larger number of hydrophobic moieties is beneficial for spontaneous insertion into membranes. Using insights gained from modified first-generation designs, we create a DNA membrane channel (termed 'T pore') following an alternative design strategy, comprising a large central pore (with a diameter of ≈ 4 nm) and a flat extramembranal part that enables a tighter attachment to the membrane. In single-molecule translocation experiments, these pores are shown to allow the passage of correspondingly larger analytes such as double-stranded (ds) DNA molecules. Moreover, the DNA channels are shown to spontaneously insert into the membranes of giant liposomes, demonstrating their potential as components for the creation of artificial cell-sized reaction compartments.

## Results

### Design of DNA membrane channels

The structure of the T pore comprises a double-layered plate ($46 \times 51$ nm, origami square lattice design[21], Supplementary Fig. 1i,j and Supplementary Note 1) with 57 hydrophobic modifications at its bottom and a rectangular aperture with dimensions $3.7 \times 8.4$ nm$^2$ in its centre (Fig. 1a,c and Supplementary Figs 2 and 5c). From the aperture, a 27 nm-long hollow stem (inner cross-section $4.2 \times 4.2$ nm$^2$) extends perpendicularly from the plate and is designed to act as a transmembrane channel. As an alternative to cholesterol, we utilized tocopherol as the hydrophobic modification, since tocopherol is known to interact well with the disordered membrane phases formed by the phosphatidylcholines used in our experiments[22,23]. The correct formation of the pore was assessed using transmission electron microscopy (TEM, Fig. 1b,d and Supplementary Fig. 3). Experiments performed in the presence of small unilamellar vesicles (SUVs) confirmed the desired tocopherol-mediated interactions of the channels with lipid bilayer membranes (Fig. 1e,f and Supplementary Fig. 4a).

In order to explore the influence of pore shape and number of hydrophobic modifications, in preliminary studies we had also created several variations of our previously published membrane channel[4] (Supplementary Fig. 1). The first design (the 'pin pore') was composed of 54 parallel double helices and a central six-helix

bundle pore, but in contrast to the initial design (containing single-stranded scaffold loops at the top of the cap), the complete scaffold was included and used for the creation of a large vestibule at the entrance of the channel. Membrane incorporation of the pore was facilitated by hybridization of the membrane-facing part of the vestibule with a total of 26 tocopherol-modified anchor strands (Supplementary Fig. 5a, cf. Supplementary Figs 6a and 7 for electron microscopic characterization). We also created a pore (termed 'wheel pore') with a central six-helix bundle pore of reduced length $l = 15.6$ nm, which had a flatter extramembranal part. This pore design could be functionalized with up to 57 hydrophobic modifications and thus allowed for stronger anchoring of the pore to the membrane (Supplementary Figs 1, 5b, 8a and 9a).

Hydrophobically modified DNA nanostructures naturally tend to aggregate. However, because of the peculiar shape of the DNA origami channels, hydrophobic moieties are never completely buried within the aggregates, and thus the channels can still interact with lipid bilayer membranes (Supplementary Figs 4a,b, 7 and 10a). Aggregation could, in principle, be reduced by the use of surfactants[24]. An alternative membrane incorporation strategy that avoids aggregation altogether is based on the use of biotin–streptavidin linkages to biotinylated lipid membranes, and is discussed further below (Supplementary Figs 3b, 4b, 6b, 8b, 9b and 10).

### Electrical characterization

In order to directly prove membrane channel formation, we performed electrical single-channel recordings with all our DNA nanopore designs (Supplementary Figs 11–13). The conductance of each pore was determined using the droplet interface bilayer (DIB) technique[25,26] (Fig. 2d), which in our hands enabled more reproducible conductance measurements than previously used methods (Methods). For the pin pore we observed a conductance of $G = 1.6$ nS (cf. Supplementary Table 1); however, the structure only rarely incorporated into the membrane, which we attributed to its relatively small number of hydrophobic modifications. By contrast, the wheel pore—with 57 hydrophobic anchors—showed improved membrane interactions, but tended to aggregate in solution. As a result, we frequently observed insertion of multiple channels at a time and a correspondingly higher apparent conductance (Supplementary Fig. 11). Analysis of the data allowed us to extract a single-channel conductance of $G = 1.5$ nS, which is consistent with the dimensions of the channel and is similar to that of the pin pore. In contrast to the other pores, the large diameter T pore displayed a much higher membrane insertion frequency, and its larger pore diameter resulted in a correspondingly higher conductance value of $G = 3.1 \pm 0.3$ nS. Control experiments with DNA nanostructures lacking a membrane stem structure did not result in any measurable transmembrane current.

In contrast to previous measurements with a planar suspended lipid bilayer set-up[4], we did not observe spontaneous channel closure (gating) when using the DIB technique, which may be caused by the different nature of the membrane in the interface bilayer. The relatively good correspondence between the conductance and the dimension of the channel (Supplementary Table 1 and Supplementary Note 2) could be the result of a reduced leak current through the structure; however, we also cannot rule out an accidental cancellation of several contributions to the overall conductance. For instance, there could be current leakage along the sidewalls of the pore structure, electro-osmotic effects and the DNA structure might be compressed within the bilayer, reducing the effective diameter of the channel. Of note in this context, membrane pore formation was recently also demon-

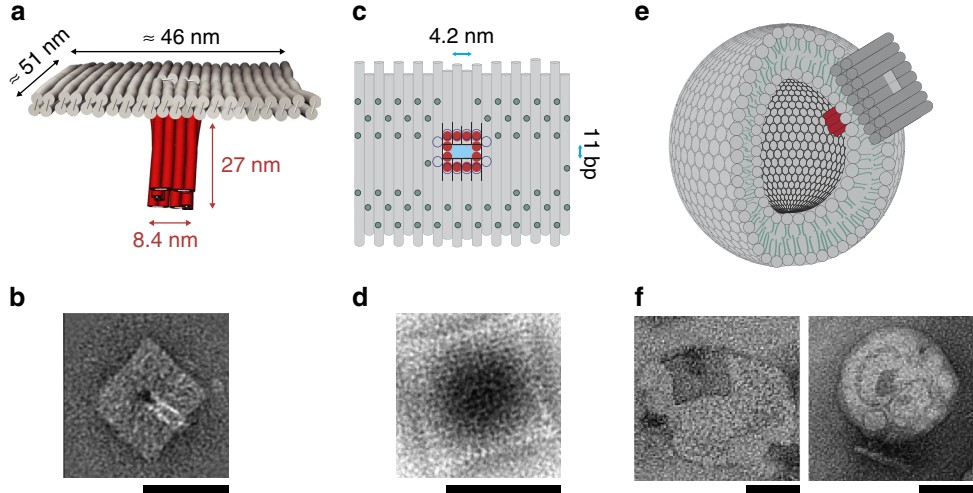

**Figure 1 | Design and electron microscopic characterization of the DNA T pore membrane channel.** (**a**) Design of the T pore (side view), which is composed of a double-layered top plate and a 27 nm long stem attached to the centre of the plate. The stem is formed by 12 helices arranged in a square (cross-section $4.2 \times 4.2 \, \text{nm}^2$). The plate provides a large area for membrane interactions. (**b**) Class-average of TEM images of the T pore showing the plate and stem of the structure. (**c**) Schematic top view of the-T pore-showing the top of a double-layered DNA plate. The green circles represent the distribution of the 57 possible positions for the introduction of membrane-interacting modifications (tocopherol or biotin). The red circles depict the stem region of the structure. The top layer (black lines in the schematic) has a hole of $3.7 \times 8.4 \, \text{nm}^2$. The bottom layer represented in dark blue in the schematic has a hole of $5.7 \times 8.4 \, \text{nm}^2$. The cross-sectional area (light blue in the schematic) of the effective pore spanning through the membrane is determined by the area of the hole in the first and second layer of the plate as well as the cross-section of the stem, which has the dimensions of 11 base pairs (bp) $\times$ 4.2 nm (that is, $3.7 \times 4.2 \, \text{nm}^2$). (**d**) TEM class-average image of a T pore plate lacking the stem structure. The unused scaffold from the stem portion is seen as a darker region on the plate. (**e**) Schematic representation of a T pore interacting with a SUV. (**f**) TEM images showing membrane interactions of tocopherol-functionalized T pores with PC SUVs. All scale bars, 50 nm.

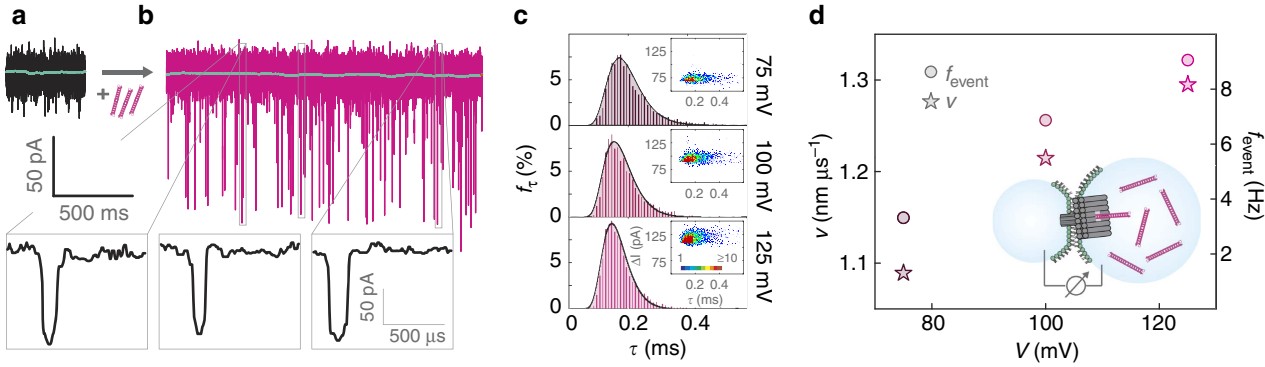

**Figure 2 | Translocation of dsDNA through large diameter T pore channels.** (**a**) Sample trace of a current measurement through a single DNA channel incorporated into a lipid bilayer membrane in the absence of analyte molecules, performed at a transmembrane voltage of 125 mV. The green line indicates the median of the current over 100 ms ($\approx 365 \, \text{pA}$). (**b**) Current trace recorded in the presence of 527 base pair long dsDNA. The green line indicates the median of the current over 100 ms ($\approx 385 \, \text{pA}$). Three examples for translocation events are shown in detail below the trace. (**c**) Distribution of event durations extracted from current traces recorded at 75, 100 and 125 mV. The times occurring with the highest probabilities (170, 154 and 149 μs at 75, 100 and 125 mV, respectively) are defined as DNA translocation times at the corresponding voltages. The histograms were fit with a one-dimensional drift-diffusion model for nanopore translocation[7] with the two fit parameters $v$ and $D$ (velocity and diffusion coefficient; *cf.* Supplementary Note 3). In the inset, a scatter plot of the current reduction $\Delta I$ versus the translocation time $\tau$ is shown for all events detected at 75, 100 and 125 mV. As expected, $\Delta I$ increases with voltage, while the event duration decreases. (**d**) Translocation velocities and event frequencies extracted from the measurements. Consistent with dsDNA translocation, both parameters increase with increasing voltage. Inset: all electrical recordings were performed with the droplet interface bilayer technique (*cf.* Methods). Lipid bilayers were formed at the interface between two electrically contacted droplets containing water, 1 M KCl and 5 mM $MgCl_2$ encapsulated by lipid monolayers made from DPhPC.

strated by single, hydrophobically tagged DNA duplexes[27]. Further studies using molecular dynamics simulations[27–29] could help to clarify these issues.

**Single-molecule translocation experiments**. DNA membrane channels can be utilized as stochastic single-molecule sensors[12,15] with tunable properties. In nanopore sensing, a constant voltage is applied across a pore-containing membrane, resulting in an ionic current through the nanopore. Electrophoretic translocation of analyte molecules leads to a reduction in channel current, whose duration and depth depend on the charge and size of the analytes[11,30]. In contrast to other DNA nanopores and typical biological membrane channels, the increased size of the T pore permitted the passage of biomolecules with larger diameter (Supplementary Fig. 14). For example, T pores allowed the translocation of 527 base-pair (bp) long ds DNA molecules

(Fig. 2a,b). As expected, the frequency of translocation events and the speed of the molecules increased with transmembrane voltage, while the dwell time in the channel decreased from 170 μs (75 mV) over 154 μs (100 mV) to 149 μs (125 mV; Fig. 2c,d). The relative reduction in conductance $\Delta G/G$ in the presence of dsDNA corresponds very well to the relative reduction in cross-sectional area of the pore, also indicating a relatively low level of leakage in our measurements (Supplementary Fig. 15).

As further shown in the Supplementary Fig. 16 and Supplementary Note 3, a single-stranded obstruction inside of the channel additionally slowed down the speed of the translocations. We also found that translocation of single-stranded DNA resulted in less-pronounced current reductions than for dsDNA analytes, consistent with the smaller diameter of ssDNA (Supplementary Fig. 16).

**Dye influx studies with giant liposomes.** In order to assess membrane incorporation in the absence of a membrane potential, we performed dye influx assays (Supplementary Note 4) using surface-immobilized giant unilamellar vesicles (GUVs) monitored using confocal laser fluorescence microscopy (Fig. 3 and Supplementary Movies 1–4). Previously, spontaneous insertion into bilayers was only demonstrated in experiments with SUVs[4,31], whose highly curved membranes are easily penetrated by DNA membrane channels. By contrast, the membranes of GUVs (with diameters of more than 10 μm) are flat on the scale of the DNA pores and are much less accessible for pore formation.

Giant liposomes are of considerable interest as models for artificial cells[20], and in contrast to SUVs allow a direct microscopic observation of DNA channel membrane interaction and insertion. As GUVs are much more delicate to handle than SUVs, GUV preparation had to be carried out carefully to minimize osmotic pressure imbalances across the membrane. In our experiments, the interior of the vesicles initially contained buffer solution only, while the exterior solution also contained the fluorescent dye Atto 633, which does not penetrate the lipid bilayer. After addition of nanopores in nanomolar concentration, the vesicles were monitored for dye influx for several hours. We found that both T pore and wheel pore were able to induce the dye influx, while the pin pore was not (Supplementary Figs 17–19), which is consistent with our electrical recordings in which the pin pore rarely incorporated into the membrane even in the presence of an electric field. In experiments with fluorescently labelled pores we observed immediate adsorption of the pores to the vesicles[26], followed by a lag time after which the membrane was perforated (Fig. 3e).

For the T pore, the dye influx usually followed exponential kinetics (Supplementary Fig. 20), and larger vesicles typically displayed faster influx rates. As shown in Fig. 3f, influx rates correlated with the surface area of the vesicles, indicating the simultaneous insertion of a number of channels, which is roughly proportional to this area. For comparison, similar experiments were performed using the protein pore alpha haemolysin (Supplementary Figs 21 and 22).

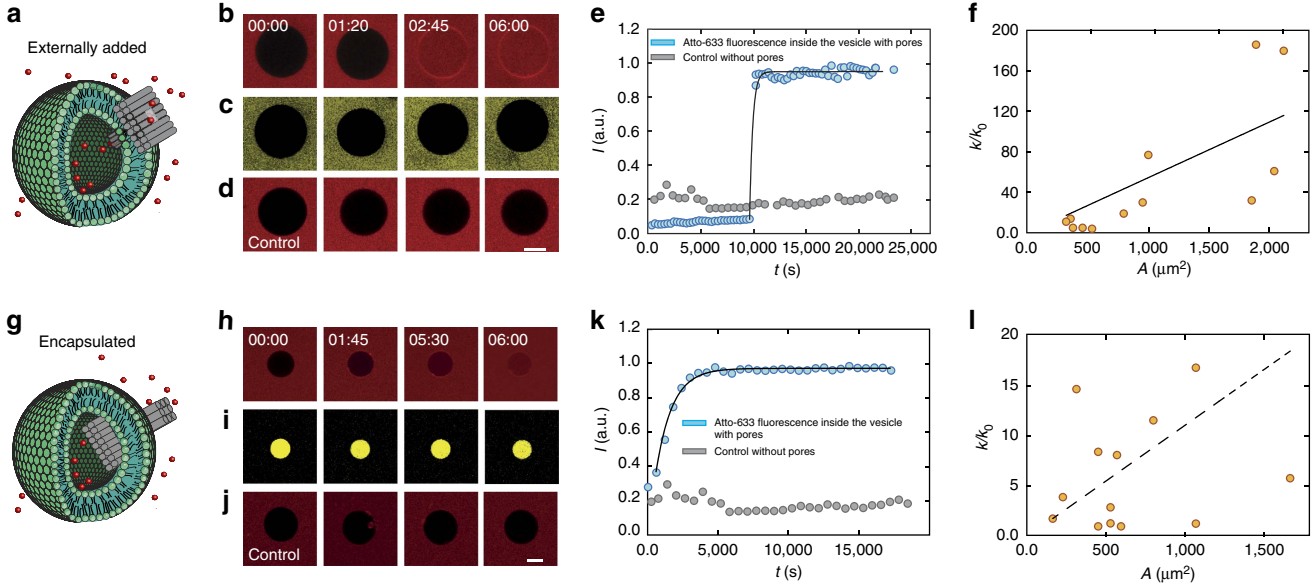

**Figure 3 | Dye influx experiments with surface-immobilized GUVs.** (**a**) Schematic of a T pore penetrating a vesicle. (**b**) Time series of confocal images recorded during a dye influx experiment monitoring the increase in fluorescence of Atto 633 inside the vesicle after the pores are added at $t = 0$ (Supplementary Movies 1 and 2). (**c**) The same vesicle as in **b** observed in the fluorescein channel. The larger fluorescein–dextran conjugate present in the outside solution is unable to enter the GUV (nominal radius of 14.7 nm). (**d**) Control experiment in the absence of DNA origami channels showing no dye influx. (**e**) Fluorescence trace (blue dots) recorded from the GUV shown in **b**. After a time delay of $t \approx 1$ h, a sudden influx of Atto 633 into the vesicle is observed, which approximately follows exponential kinetics (Supplementary Fig. 20). (**f**) The influx rate $k$ is estimated by fitting the fluorescence intensity with an exponential model for dye influx (Supplementary Note 4). The influx rate $k$ observed for each vesicle is normalized by dividing it with the expected rate for a single pore $k_0$ and plotted as a function of the surface area, $A$, of the corresponding vesicles. The line is obtained from linear regression analysis. (**g**) Schematic of a T pore incorporated into a vesicle in an inside-out configuration. Vesicles were loaded with origami solution containing a dextran–fluorescein conjugate as described in Supplementary Note 4. (**h**) Confocal fluorescence images monitoring the pore-mediated influx of Atto 633. (**i**) Time series monitoring the same vesicle as in **h** in the fluorescein channel, demonstrating that in contrast to Atto 633 the dextran conjugate cannot leak out. (**j**) Control experiment in the absence of DNA origami channels showing no dye influx (Supplementary Movies 3 and 4). (**k**) Atto 633 fluorescence time trace corresponding to the GUV shown in **h**. As above, the influx follows exponential kinetics, but as encapsulation is carried out before confocal imaging, typically a reduced or no time delay is observed. (**l**) $k/k_0$ versus $A$ from vesicles with encapsulated pores. $k/k_0$ is smaller and the correlation between $k/k_0$ and $A$ is weaker than for externally added pores. Scale bars, 10 μm.

Control experiments in the absence of pores were performed to rule out accidental bursting or leaking through spontaneously formed membrane pores under our experimental conditions (Fig. 3d). Dye molecules tethered to a dextran moiety larger than the pore diameter were unable to enter the vesicles (Fig. 3c).

We also studied the dye influx through DNA channels inserted into the membrane from the interior of the vesicles in an 'inside-out' configuration (Fig. 3g and Supplementary Note 5). To this end, we encapsulated DNA channels using an inverted emulsion technique[32] (Methods, Supplementary Fig. 23), which also resulted in a successful perforation of the membrane (Fig. 3h). Typically, the influx of dye occurred more slowly in these experiments, corresponding to a smaller number of simultaneously inserted pores (Fig. 3k,l). As also shown in Fig. 3i, while Atto 633 dye entered through the pores into the vesicles, encapsulated dye–dextran conjugates did not leak out.

**Streptavidin-mediated membrane interactions**. Finally, we investigated an alternative strategy for inserting and anchoring DNA nanochannels into lipid membranes, which utilizes biotinylated lipid molecules (Fig. 4a and Supplementary Movie 5). In place of hydrophobic channel modifications we attached biotinylated anchor strands, which interacted with the biotinylated lipids via streptavidin 'bridges' which allowed specific membrane interactions of the pore using a protein-ligand interaction strategy

(Fig. 4b–d). Pore structure and membrane interactions were checked using electron microscopy (Supplementary Figs 3b, 4b, 6b, 8b, 9b and 10). As shown in Fig. 4e, this strategy also successfully led to membrane perforation (for electrical characterization see Supplementary Fig. 13). Incorporation kinetics appeared to be slower than that for the hydrophobic approach (Fig. 4f), apparently resulting in the incorporation of smaller numbers of membrane pores (Fig. 4g). This is consistent with the assumption that in the absence of channel aggregation DNA pores interact with the membranes individually.

## Discussion

The shape and size of DNA-based nanochannels can be easily adjusted using the tools of DNA nanotechnology. They can be modified with multiple chemical functionalities, at exact stoichiometry and with nanometre precision, which allows tuning of the channels to meet the requirements of specific applications. We have here demonstrated the construction of a large-diameter DNA channel, which is composed of a rectangular plate as the membrane-interacting region, and a central stem perpendicular to the plate with inner dimensions of nominally $4 \times 4$ nm. Using a droplet-interface bilayer set-up for electrical single-channel recordings, the ionic conductance of the DNA channel was shown to be ohmic and with $G \approx 3$ nS in good agreement with its size. Of particular note, the large diameter of the channel allowed for

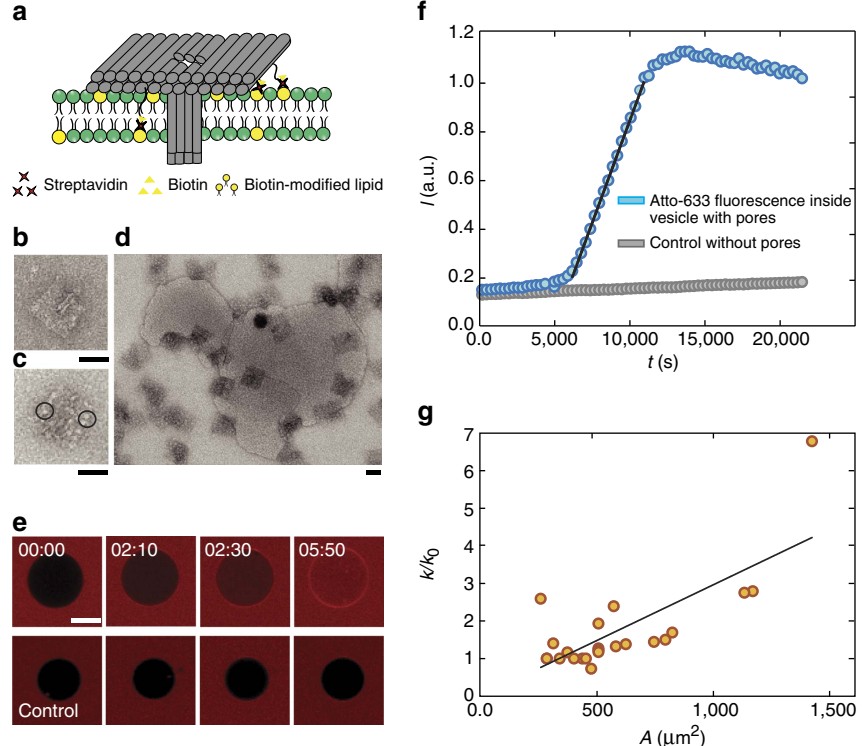

**Figure 4 | Membrane penetration mediated by biotin–streptavidin linkages.** (**a**) An alternative strategy to mediate DNA nanopore–membrane interactions based on streptavidin linkages between biotinylated staple strands at the bottom of the the T pore plate and biotinylated lipids. (**b**) TEM image of a non-functionalized T pore. (**c**) TEM image of a biotinylated T pore decorated with streptavidin. Two streptavidin proteins are highlighted in black circles. (**d**) TEM image of biotinylated T pores interacting with SUVs (90% Egg phosphatidylcholine, 10% biotinylated 1,2-dioleoyl-sn-glycero-3-phosphoethanolamine-N-(biotinyl) (sodium salt) (DOPE) in the presence of streptavidin. All TEM images have a scale bar of 50 nm). (**e**) Time series of fluorescence images of an immobilized GUV obtained from a dye influx experiment using confocal microscopy as described in Fig. 3. At $t = 0$, biotinylated T pores are added to the solution (streptavidin already present on T pores, which can be seen in image **c**; cf. Supplementary Movie 5). Dye influx demonstrates successful penetration of the membrane, which is absent in control experiments without pores. Scale bar, 10 μm. (**f**) Fluorescence intensity inside of the vesicles in **e**. Blue circles: pore formation by biotinylated T pores, grey circles: control. Data are fit with the linearized influx model (cf. Supplementary Note 4). (**g**) Normalized influx rate $k/k_0$ plotted against the area $A$ of the vesicles. Lack of aggregation is translated to slower influx rates compared with the hydrophobically modified pores.

electrically driven translocation of dsDNA, which previously was only achieved with solid-state nanopores[10] or engineered biological pores[33]. Earlier experiments with solid-state nanopores modified with DNA origami adaptors with a comparable aperture[7] as the T pore displayed a similar conductance reduction upon dsDNA translocation. Owing to the excessive leak current in these experiments, however, the relative conductance change was only on the order of ≈1%—in contrast to the $\Delta G/G$ of ≈30% measured here.

Large-diameter DNA nanopores have a wide range of potential single-molecule biosensor applications such as protein sensing or the characterization of protein–DNA and protein–protein interactions. Larger channels than the T pore could be realized based on the same construction principle, limited only by the availability of a sufficient number of membrane-anchoring positions and the available origami scaffold length. Alternatively, channels could be constructed from multiple subunits, and also structures comprising multiple parallel channels could be realized.

The flat membrane-interacting top plate of the T pore with its large number of hydrophobic modifications considerably improved binding of the structure to lipid bilayer membranes. This even facilitated the spontaneous insertion of DNA channels into self-assembled GUVs in the absence of a transmembrane voltage, which could be directly monitored in confocal laser microscopy experiments. In contrast to previous, more indirect characterization studies, microscopic observation of the dye influx into the vesicles over long periods of time also helped to confirm membrane intactness and size-specific pore formation. Finally, our alternative use of biotin–streptavidin membrane linkages demonstrates that membrane incorporation of DNA origami pores can also be mediated via protein–ligand interactions.

Our experiments on molecular transport through DNA membrane channels point towards a wide variety of applications beyond single-molecule nanopore sensing. For instance, DNA pores could act as programmable nanomedical agents that penetrate cell membranes in a context-dependent manner, and they could be used as precisely tailored channels and transporters for the creation of artificial cellular reaction compartments.

## Methods

**Fabrication of DNA nanopores.** DNA nanopores were fabricated in a one-pot reaction by stepwise cooling of a mixture of staple and scaffold strands (50 nM) in folding buffer (1 × Tris-acetate-EDTA buffer, 20 mM MgCl$_2$) from 65 to 30 °C as described previously[34]. Staple strands were designed using the caDNAno software[9] (Supplementary Fig. 24a–c and Supplementary Data 1). In order to minimize the number of unused nucleotides in the scaffold, each pore was synthesized with a different scaffold length[35]. The pin pore was synthesized using a scaffold of length 7,249 nt, for the T pore a 7,560 nt and for the wheel an 8,064 nt-long scaffold was used. The scaffolds used are modified versions of the single-stranded M13 phage genome. Functionalized DNA pores with proximally positioned[4,36] tocopherol were produced by incubating the fully folded pores with tocopherol-modified strands (Biomers) for 45 min. Before incubation, tocopherol-modified oligonucleotides were heated to 60 °C for 45 min to avoid aggregation. The folded origami pores contained single-stranded extensions complementary to the modified strands. Excess staples were removed by multiple rounds of filter purification using 100 kDa Amicon (Millipore) filters. For experiments using streptavidin bridges, DNA channels were incubated with biotinylated staple strands, which were further functionalized with streptavidin. Free protein was removed via several rounds of filter purification. From ultraviolet absorbance measurements, the final concentration of the DNA channels is estimated to be ∼80 nM after purification and concentration.

**Transmission electron microscopy.** TEM images of DNA nanostructures were recorded using a standard negative staining protocol. The TEM assay establishing DNA nanopore–membrane interactions using SUVs was carried out as described previously[4]. Vesicles and nanostructures were incubated in solution and this mixture was applied to the TEM grids, followed by negative staining. Fully intact SUVs are not filled with stain and appear white against the background, whereas DNA pores generate higher contrast. GUVs with encapsulated pores were also imaged in TEM using negative staining. Owing to their large size, GUVs burst on the grid and the bilayer allows stain to pass. DNA nanostructures again retain more stain compared with the bilayer and appear darker above than when below a burst vesicle.

**Electrical measurements.** Electrical measurements were performed using a DIB set-up, which has been described previously[25]. Briefly, aqueous droplets (one with a diameter of ≈40 μm and a large reservoir droplet with diameter 0.5 mm) were attached on two electrodes, which reside in a lipid–oil mixture (1,2-diphytanoyl-sn-glycero-3-phosphocholine dissolved in hexadecane at a concentration of 10 mg ml$^{-1}$). This led to the formation of a monolayer around the droplets. Positioning the droplets in close proximity resulted in stable bilayer formation. DNA nanopores were added to one of the droplets, and a transmembrane electric field was applied to increase incorporation probability.

**DNA translocation experiments with T pores.** Buffer conditions used for translocation measurements were 1 M KCl, 5 mM MgCl$_2$. dsDNA (527 bp) was introduced to the *cis* side of the T pore. The electrical current, measured with a patch clamp amplifier (EPC9, HEKA Elektronik GmbH), was digitized at 200 kHz. An additional gliding median filter, averaging over 20 points, was applied before current blockade analysis. As a result, the minimum detectable duration of a translocation event was 50 μs, and event durations above 5 ms were rare and not considered in our analysis. Analysis was performed using custom Matlab routines. The scatter plots shown in Fig. 2c contain $n = 2,865$ (75 mV), 5,907 (100 mV) and 7,278 (125 mV) events.

**Production of SUVs and GUVs.** All lipids used in this study were obtained from Avanti Polar Lipids. SUVs used in TEM imaging to confirm membrane interaction of the pores were created as described previously[4]. Briefly, 1-palmitoyl-2-oleoyl-sn-glycero-3-phosphocholine (POPC) lipids were dissolved in chloroform to a concentration of 5 mg ml$^{-1}$. A lipid film was formed by evaporating 1 ml of POPC solution in a 5 ml round-bottom flask, using a rotational evaporator. The lipid film was resuspended in 1 ml of a 1 M aqueous KCl solution. Subsequently, SUVs were formed by sonicating the solution with a tip sonicator (Bandelin Sonoplus mini20) for *ca.* 5 min at 8 W, until the solution started to become transparent. GUVs used in dye influx assays were produced using the inverted emulsion technique[32]. Briefly, a uniform lipid film containing 90% Egg PC or POPC lipids and 10% biotinylated 1,2-dioleoyl-sn-glycero-3-phosphoethanolamine-N-(biotinyl) (sodium salt) was created using a rotary evaporator. The lipid film was dissolved in mineral oil by sonication and vortexing. In order to form the GUVs, 20 μl of the solution to be encapsulated inside the vesicles (the internal solution (IS), containing 400 mM sucrose) was added to the lipid–oil mixture and vortexed, which resulted in the formation of droplets surrounded by a lipid monolayer. This emulsion was carefully layered on top of an external solution (ES). The ES had the same osmolarity as the IS to ensure stable GUV formation, but contained 400 mM glucose. GUVs were then formed by centrifuging the monolayer droplets through the IS–ES interface. When origami pores were added externally to the vesicles, the ES and IS were adjusted to a final salt concentration of 500 mM KCl to preserve the structure of the origami pores. Encapsulation of DNA origami structures was carried out by replacing the IS containing only sucrose to a mixture of DNA origami and sucrose, while the final salt concentration of the IS was maintained at 500 mM KCl. The ES was supplemented with salt and glucose to equalize the osmolarity on both sides of the vesicle membrane. Encapsulation with dextran (MW 500,000)–fluorescein conjugate (Sigma-Aldrich) was carried out similarly (Supplementary Notes 4 and 5).

**Dye influx assay and confocal microscopy.** The dye influx assay was carried out by immobilizing GUVs containing 10% biotinylated lipids on a BSA–biotin–streptavidin-coated ibidi chamber. Origami structures were added externally or encapsulated in the vesicles as described above. Modifications mediating interactions with the lipid bilayer were added to the origami structure during purification. The dye influx assay was carried out at room temperature. Influx of dye was observed using a confocal microscope (Leica SP5 II).

**Experiments with streptavidin-functionalized DNA channels.** For experiments with streptavidin-functionalized DNA channels, vesicles were prepared with 10% biotinylated lipids with sucrose inside the vesicle. In order to be able to utilize these lipids for channel incorporation, vesicles were not immobilized on BSA–biotin slides. Instead, DNA channels were simply added to sucrose-containing vesicles in a glucose-containing ES. Owing to the density difference, the vesicles settled on the surface of the microscope slide, allowing observation of lipid vesicles over a sufficient period of time.

**Data availability.** The data that support the findings of this study are available from the corresponding author upon request.

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

## Acknowledgements

We gratefully acknowledge the financial support through the DFG (SFB 863 TP A8, A9, B1 and Nanosystems Initiative Munich). We also thank Ali Aghebat Rafat and Klaus Wagenbauer for useful discussions, and Florian Praetorius for kindly providing origami scaffold.

## Author contributions

S.K., D.Z., V.A., T.M., A.R.B., H.D. and F.C.S. planned and designed the experiments; S.K., D.Z., V.A. and K.H. performed the experiments; S.K., D.Z., V.A., K.K. and F.C.S. analysed the data; S.K., D.Z., V.A. and F.C.S. wrote the paper; all authors discussed the results and commented on the manuscript.

## Additional information

**Competing financial interests:** The authors declare no competing financial interests.

