## [Peer Review File · Nature Communications]

Transferred manuscripts:

NCOMMS-16-12317-T

REVIEWERS' COMMENTS:

Reviewer #1 (Remarks to the Author):

The authors have substantially improved the manuscript to address all comments I raised in the previous review. I am happy to support publication without further changes.

The focus on the single improved origami design, and the voltage-dependence of translocation speed helps satisfy my initial concerns that this pore was functioning 'as designed'. In particular highlighting the significant magnitude of the current blockades relative to open-pore currents helps emphasize the advance.

My only suggestion would be to include a comment to the effect that they attempted PEG-sizing of the pore but were unsuccessful.

Reviewer #2 (Remarks to the Author):

I am reviewer #2 for the first version of this manuscript (submitted to Nature Nanotechnology). I continue to hold my view that this work is novel (structurally rigid DNA nanopore, detergent-free insertion to GUVs, and DNA translocation), thorough (well documented statistics and analyses), and potentially impactful (applications in cell-size reactor). All my previous concerns have been adequately addressed. Overall, I feel this revised manuscript is ready to publish in Nature Communications. I have only one request: could the authors PLEASE include cross-section views with helix numbers together with the caDNAno designs in SI for the benefit of non-specialists?

Reviewer #3 (Remarks to the Author):

The manuscript by Simmel et al. is a resubmission of a revised manuscript. After reading the comments, responses and manuscript, I am convinced that the manuscript is suitable for publication. The responses are clearly laid out and address the concerns of the referees.

As a note, the authors describe in the manuscript that other mechanisms of additional current rather than through the pore cannot be ruled out. In this respect, the authors need to discuss a recent paper by Stulz, Aksimentiev and Keyser (Nano Lett 2016) and add some comments on the findings on current formation alongside simple DNA pores; this is important in view of this paper's mentioning that leakage along the sidewalls of the pore structure could be significant.

Reply to the reviewer's comments:

We thank all reviewers for their very positive assessment of our revised manuscript.

* Reviewer #1

Comment:

The authors have substantially improved the manuscript to address all comments I raised in the previous review. I am happy to support publication without further changes.

The focus on the single improved origami design, and the voltage-dependence of translocation speed helps satisfy my initial concerns that this pore was functioning 'as designed'. In particular highlighting the significant magnitude of the current blockades relative to open-pore currents helps emphasize the advance.

My only suggestion would be to include a comment to the effect that they attempted PEG-sizing of the pore but were unsuccessful.

Reply:

We included the suggestion adding a comment on unsuccessful PEG-sizing in the Supplementary Information (page 26)

* Reviewer #2

Comment:

I am reviewer #2 for the first version of this manuscript (submitted to Nature Nanotechnology). I continue to hold my view that this work is novel (structurally rigid DNA nanopore, detergent-free insertion to GUVs, and DNA translocation), thorough (well documented statistics and analyses), and potentially impactful (applications in cell-size reactor). All my previous concerns have been adequately addressed. Overall, I feel this revised manuscript is ready to publish in Nature Communications. I have only one request: could the authors PLEASE include cross-section views with helix numbers together with the caDNAo designs in SI for the benefit of non-specialists?

Reply:

We included the cross-section views as requested for the three origami pore designs in the Supplementary Information (page 19, 20, 21)

* Reviewer #3

Comment:

The manuscript by Simmel et al. is a resubmission of a revised manuscript. After reading the comments, responses and manuscript, I am convinced that the manuscript is suitable for publication. The responses are clearly laid out and address the concerns of the referees.

As a note, the authors describe in the manuscript that other mechanisms of additional current rather than through the pore cannot be ruled out. In this respect, the authors need to discuss a recent paper by Stulz, Aksimentiev and Keyser (Nano Lett 2016) and add some comments on the findings on current formation alongside simple DNA pores; this is important in view of this paper's mentioning that leakage along the sidewalls of the pore structure could be significant. add some comments on the findings on current formation alongside simple DNA pores “

Reply:

We cite the paper and discuss it in the Main text (page 5) and Supplementary Information (page 29).